



# Compilation of Last Interglacial (Marine Isotope Stage 5e) sea level indicators in the Bahamas, Turks and Caicos, and the east coast of Florida, USA

5    Andrea Dutton[1], Alexandra Villa[1], Peter M. Chutcharavan[1]

[1]Department of Geoscience, University of Wisconsin-Madison, Madison, 53706, USA

*Correspondence to*: Andrea Dutton (dutton3@wisc.edu)

**Abstract.** This paper provides a summary of published sea level archives representing the past position of sea level during the
10   Last Interglacial sea level highstand in the Bahamas, Turks and Caicos, and the eastern (Atlantic) coast of Florida, USA. These
data were assembled as part of a community effort to build the World Atlas of Last Interglacial Shorelines (WALIS) database.
Shallow marine deposits from this sea level highstand are widespread across the region and are dominated by carbonate
sedimentary features. In addition to depositional (constructional) sedimentary indicators of past sea level position, there is
also evidence of erosion, dissolution, and/or subaerial exposure in places that can place an upper limit on the position of sea
15   level. The sea level indicators that have been observed within this region and attributed to Marine Isotope Stage (MIS) 5e
include corals, oolites, and other coastal sedimentary features.

Here we compile a total of 50 relative sea level indicators including 36 in the Bahamas, three in West Caicos, and a remaining
10 for the eastern seaboard of Florida. We have also compiled U-Th age data for 24 fossil corals and 56 oolite samples. While
20   some of these archives have been dated using U-Th disequilibrium methods, amino acid racemization, or optically stimulated
luminescence, other features have more uncertain ages that have been deduced in the context of regional mapping and
stratigraphy. Sedimentary archives in this region that constrain the elevation of the past position of sea level are associated
with uncertainties that range from a couple decimeters to several meters. Across the Bahamas and on West Caicos, one of the
observations that emerges from this compilation is that estimation of sea level position in this region during Marine Isotope
25   Stage 5e is complicated by widespread stratigraphic evidence for at least one sea level oscillation. This evidence is defined by
submarine features separated by erosion and subaerial exposure, meaning that there were likely multiple distinct peaks in sea
level rather than just one. To this end, the timing of these individual sea level indicators becomes important when compiling
and comparing data across the region given that different archives may have formed during different sub-orbital peaks in sea
level.

## 1 Introduction

The goal of this data compilation is to synthesize existing data for sea level position during the Last Interglacial across a region that includes the Bahamas, Turks and Caicos, and the eastern coast of Florida. This contribution is part of a community effort to produce a global database of Last Interglacial sea level indicators that is open-access and meets FAIR (Findable, Accessible, Interoperable, and Reusable) data standards (Wilkinson et al., 2016). Here we adopt the standardized framework provided by the World Atlas of Last Interglacial Shorelines Database (WALIS, https://warmcoasts.eu/world-atlas.html) to compile a database of previously published relative sea level (RSL) indicators including their descriptions, geochronological constraints, and associated metadata.

In this contribution to the WALIS database, our focus is on Marine Isotope Stage (MIS) 5e rather than MIS 5 as a whole because a global compilation of sea level indicators from MIS 5a and 5c are included in Thompson and Creveling (2021). We also only include the east coast of Florida (including the Florida Keys) in this regional compilation; complementary data for the west coast are included in the compilation of Simms (2021). Given the widespread nature of shallow marine deposits from MIS 5e across this region, we focus our data compilation to the best constrained sea-level indicators in the literature in terms of both age and elevation.

## 2 Sea level indicator types and uncertainties

There are several types of sea level indicators that we have compiled for the WALIS database from the Bahamas-Turks and Caicos-Florida region. Because of the dominance of carbonate substrate in this region, most of these are constructional or erosional (including dissolution) features associated with carbonate sedimentary environments.

The most common methods used to directly date these features are amino acid racemization (AAR) or U-series disequilibrium dating. For each entry into the WALIS database, we have specified the dating technique, which in some cases relies on the stratigraphy relative to dated features above or below. We have not included features assumed to be MIS 5e in the absence of other supporting evidence for the age assignment.

Both the age and elevation of the sea level indicators that we have entered into the WALIS database are assigned a subjective quality rating using the definitions provided in Tables 1 and 2 to help database users assess the uncertainties associated with both parameters.

**Table 1.** Quality assessment of the relative sea level data in terms of vertical uncertainty, as defined in the WALIS database documentation: https://walis-help.readthedocs.io/en/latest/RSL_data.html#quality.





| Quality Rating | Description |
|---|---|
| 5 | Elevation precisely measured, referred to a clear datum and RSL indicator with a very narrow indicative range. Final RSL uncertainty is submetric. |
| 4 | Elevation precisely measured, referred to a clear datum and RSL indicator with a narrow indicative range. Final RSL uncertainty is between one and two meters. |
| 3 | Uncertainties in elevation, datum or indicative range sum up to a value between two and three meters. |
| 2 | Final paleo RSL uncertainty is higher than three meters |
| 1 | Elevation and / or indicative range must be regarded as very uncertain due to poor measurement / description / RSL indicator quality |
| 0 (rejected) | There is not enough information to accept the record as a valid RSL indicator (*e.g.,* marine or terrestial limiting) |

**Table 2.** Quality assessment of the relative sea level data in terms of age uncertainty, as defined in the WALIS database documentation: https://walis-help.readthedocs.io/en/latest/RSL_data.html#quality.

| Quality rating | Description |
|---|---|
| 5 | Very narrow age range, *e.g.*, a few ka, that allow the attribution to a specific timing within a substage of MIS 5 (e.*g.,* 117±2 ka) |
| 4 | Narrow age range, allowing the attribution to a specific substage of MIS 5 (*e.g.,* MIS 5e) |
| 3 | The RSL data point can be attributed only to a generic interglacial (*e.g.,* MIS 5) |
| 2 | Only partial information or minimum age constraints are available |
| 1 | Different age constraints point to different interglacials |
| 0 (rejected) | Not enough information to attribute the RSL data point to any Pleistocene interglacial. |

We briefly describe the different categories of sea level indicators below along with their relation to past sea level position. This includes classification as marine-limiting (below the position of sea level), terrestrial-limiting (above sea level), or direct sea level markers (known position close to sea level). In the case of direct sea level markers, it is necessary to define the indicative meaning of the sea-level indicator, which refers to the position of the sea-level indicator relative to the concomitant position of sea level. The WALIS database uses the concept of indicative meaning to define the elevation of a where a particular sea level indicator forms relative to the position of a tidal datum, usually mean sea level. The concept of indicative meaning was developed in the context of studies of Holocene sea level to provide a systematic way of using modern analogues to help reconstruct past sea level (Van De Plassche, 1986; Shennan et al., 2015). To do this, for each direct sea level indicator we have entered in the WALIS database we report the elevation of the indicator as measured relative to a modern tidal datum,





along with the indicative range (IR) of the indicator. The IR is the range refers to the elevation range where an indicator forms relative to a tidal datum. This helps to define the reference water level (RWL), which is the mid-point of the IR. These terms are summarized in Table 3 and illustrated with an example in the description of the fossil coral indicator, below.


**Table 3.** Indicator types included in this compilation (see text for detailed descriptions).

| Indicator Name | Indicator type | Reference water level (RWL) (m) | Indicative Range (IR) (m) | Comments |
|---|---|---|---|---|
| Coral patch reef | direct | -3 | 0 to -6 | In some cases, additional information was used to further constrain the paleowater depth position to a narrower range as described in the text. |
| Coralline red algae | direct | -0.4 | -0.6 to -0.2 | |
| Exposure horizons | terrestrial-limiting | above sea level | | |
| Flank margin caves | direct | poorly constrained | poorly constrained | Not included as entries in database due to uncertainties in determining RWL, IR, and age of formation. |
| Keystone vugs | direct | 0 | -0.4 to +0.4 | Adopted elevation of lowermost occurrence of keystone vugs to identify sea level position |
| Beach to dune transition | direct | 1 | -1 to +1 | Accuracy and precision of RWL and IR are estimated and may be variable across the region of study. |
| Oolite shoals | direct | -1 | -2 to 0 | The highest occurrence of ooid shoals in the area was entered in the database. |

## 2.1 Fossil Corals (direct indicator)

At a minimum, corals growing in the Bahamas, Turks and Caicos, and Florida can be considered as marine-limiting (*i.e.,* below

sea level) and depending on the level to which additional information is available regarding the paleowater depth, they may provide a more precise estimate of past sea level position. Taxonomic identification, further identification of associated reef biota (assemblage), coral morphology, and sedimentary context (facies and other features) can help to refine the paleowater depth estimate of fossil corals (e.g., Hibbert et al., 2016). Most of the MIS 5e fossil coral outcrops in this region are patch reefs that are consistent with a lagoonal setting that has often been interpreted to grow up to a position of 3 meters below sea

level (Chen et al., 1991a). However, patch reefs in the Bahamas have also commonly been observed to grow up to mean low tide (e.g., Hattin and Warren, 1989) so it is possible that some of these patch reefs were somewhat shallower or deeper than the commonly cited indicative meaning. Here we adopt an estimated paleowater depth (or RWL) of -3 m relative to mean sea level, with a possible depth range (IR) of 0 to -6 m for patch reefs in this region unless there is more specific information available that can be used to refine this range. As an example, if we measure a patch reef surface at an elevation of +1 m

relative to mean sea level, using the IR and RWL defined above, we would interpret a paleo sea level position of +4 ± 3 m (*n.b.,* this cited uncertainty only includes the uncertainty introduced by the indicative range, but the WALIS database also incorporates the measurement error of the elevation of the patch reef by adding these uncertainties in quadrature (as inRovere

et al., 2016). We note though, that the application of the indicative meaning concept inherently sets the interpreted position of sea level at the mid-point of the IR (*i.e.*, at the RWL). However, in many cases the authors have measured the elevation of the

highest parts of the reef that may represent something closer to mean sea level, in other words at the top of the IR. Additionally, we note that corals are marine limiting, meaning that sea level must have been above them, so even if the database computes an uncertainty that extends below the reported elevation of the coral, sea level must have been above the surface of the coral. Because asymmetrical uncertainties are not possible within the framework of the indicative meaning concept, this nuance is not captured in the database but can be accounted for by noting the elevation of the indicator as reported.

**2.2 Coralline Red Algae (direct indicator)**

The presence of coralline red algae caps to the coral reefs provides an additional constraint on sea level position given the narrower IR of *Neogoniolithon* sp. and *Goniolithon* sp. in modern environments. Hattin and Warren (1989) observed these to grow in a narrow range withing the uppermost subtidal to intertidal zone in the Bahamas, estimated at -0.4 ± 0.2 m. Due to this narrow indicative range, this is the most precise sea-level indicator of those we have compiled for this region.

**2.3 Exposure horizons (terrestrial-limiting)**

Subaerial exposure horizons are a common feature on carbonate surfaces when sea level falls. Such horizons can include paleosols, calcretes, caliche, and/or other diagnostic features. Exposure horizons are particularly useful to reconstruct the past position of sea level when they separate subtidal facies occurring above and below the exposure surface. In the Bahama-Florida region, *terra rossa* paleosols often develop during prolonged episodes of subaerial exposure during a glacial or

interstadial period. The origin of these reddish to orange-brown colored paleosols has been attributed to dust that is windblown into this region across the Atlantic from Africa (e.g., Muhs et al., 2007). Weakly-developed calcarenite protosols are thought to represent pauses in accumulation, possibly during periods when the carbonate banks are at least partially flooded, in contrast to major breaks between separate sea level highstands that are represented by *terra rossa* paleosols (Harmon et al., 1983). While subaerial horizons are important indicators of relative sea level position, single paleosols may bifurcate in weathered

bedrock or may merge laterally, and also can fill pits and dissolution features. Therefore, these features should be carefully interpreted within a wider sedimentary context. Other evidence that can be used to indicate subaerial exposure includes the formation of calcretes or the presence of a bored surface.

**2.4 Flank Margin Caves (direct indicator)**

Many flank margin caves are found between 1 to 7 m in elevation across the Bahamas. These have been presumed to have

formed at the distal margin of the freshwater lens during the Last Interglacial on the basis of their observed elevation, although there are some occurrences of higher flank margin caves in the region that are likely older (Mylroie et al., 2020). Because flank margin caves are dissolution features, they cannot be directly dated, but if they contain speleothems that formed within the cave, ages of speleothem growth can help to define minimum ages of cave formation. However, it has been noted that

erosion and retreat of cave openings can lead to ambiguous inferences of cave formation and corresponding interpretations of
paleo sea level position (Mylroie et al., 2020). A recent compilation of elevations of flank margin caves across the region
identified numerous caves above 7 m, reaching up to 24 m within Pleistocene host rock. This challenges the long-held notion
of a uniform 1-2 m subsidence rate for the platform, questioning the accuracy of flank margin caves as indicators of sea level
(Mylroie et al., 2020). Given the uncertainty surrounding the inference of sea level position and the challenge in ascertaining
the timing of cave formation, we have not included flank margin caves into the WALIS database.

## 2.5 Keystone vugs and Beach-to-dune transition (direct indicator)

Carbonate sand beaches can preserve fenestrae or keystone vugs that form in the swash zone and are accompanied by low-
angle bedding. Dunham (1970) coined the term "keystone vugs" to describe the occurrence of voids in grainstone that were
hypothesized to form in the swash zone. As air bubbles lift grains, they form into the shape of a keystone arch that remains
after the bubble is gone. The elevation uncertainty corresponding to the upper swash zone of the beach deposits coincides
with the local tidal range. As beach deposition can occur above the maximum tidal range during storms, the lowermost
occurrence of keystone vugs is typically taken as the best estimate for the position of sea level within the beach facies and is,
perhaps, the most accurate and precise marker of sea level across this entire region. This feature is typically interpreted as
representing the position of mean sea level, with an uncertainty of ± ½ of the local tidal range (e.g., Mauz et al., 2015).

Where keystone vugs are not identified, some authors have used the contact between beach and dune facies to define the
position of sea level. This transition is likely to occur near the position of the ordinary berm, which in turn is a function of
wave climate and sediment size. The elevation of this transition may be higher due to storm swash and depending on the wave
energy of along that particular coastline (e.g., Rovere et al., 2016). Here, we have used a nominal position of +1 m with a
range of ± 1 for the elevation of the beach-to-dune transition, noting that both the accuracy and precision of this interpretation
are uncertain.

## 2.6 Oolite shoals (direct indicator)

Though there is some topography along the surface of oolite shoals, here we have kept with the convention used by Muhs et
al. (2011) in assigning a paleowater depth of -1 m (where we assign an uncertainty of ± 1) to the highest elevation of the MIS
5e oolites in this region. Observations of modern ooid shoals indicate that sediments dominated by ooids (*i.e.*, 80-100%) on
the Bahama platform are typically found at depths ranging from 0 to 3.3 m below modern sea level with an average position
of approximately -1.2 m (Newell et al., 1960).

## 3 Bahamas sea level indicators

MIS 5e Bahamian stratigraphy is typically summarized as being represented by the Grotto Beach Formation, which was first
identified on San Salvador Island and comprises two members: the French Bay and Cockburn Town members (Titus, 1980;



Carew and Mylroie, 1985; Kindler, 2010).  The French Bay Member is composed of a reasonably well-preserved oolite exhibiting a range of environments from subtidal to intertidal to eolian (Carew and Mylroie, 1985; Caputo, 1995; Kindler and Hine, 2008).  The French Bay Member has been attributed to the early part of MIS 5e based on its stratigraphic position below the radiometrically-dated Cockburn Town Member and on the basis of amino acid racemization data that suggests a MIS 5e age (Hearty and Kaufman, 2000).  This lower member has been interpreted as representing a transgressive eolianite that

supersedes the deposition of the subtidal Cockburn Town Member, or as possibly representing an early highstand during MIS 5e (Hearty and Kindler, 1993; Kindler and Hearty, 1996; Hearty and Kaufman, 2000).  The overlying Cockburn Town Member mainly comprises coral reef boundstone or rudstone and shell-rich floatstone representing subtidal lagoonal and reef settings. This is the most common MIS 5e unit observed across the Bahamas, outcropping on many islands such as the Exumas, Great Inagua, Mayaguana, and New Providence Island.  The Cockburn Town member has been described as containing a prominent

unconformity separating two reef units.  This two-phase depositional model during MIS 5e is also supported by distinct ages within the MIS 5e stratigraphy from amino acid racemization data and U-series data (Hearty and Kaufman, 2000; Skrivanek et al., 2018).  At some localities, such as the type section at Grotto Beach on San Salvador Island, a shallowing upwards sequence is observed from subtidal reef and lagoonal facies to subaerial oolitic calcarenites.  This upper subaerial facies has been variously interpreted as the regressive phase of the Cockburn Town Member or representing a separate, late MIS 5e

highstand.

We entered 39 sea level indicators into the WALIS database that come from ten different sites across the Bahamas platform, as shown on the map in **Figure 1**.  The position of past sea level during MIS 5e across the Bahamas and Turks and Caicos region is summarized in **Figure 2**.  The following descriptions are provided to describe the stratigraphic and depositional

context of these sea level markers along with the uncertainties in elevation and age.  Most of the sites demonstrate an early MIS 5e depositional phase and a later phase, often stratigraphically separated by an erosional unconformity and/or subaerial exposure.  The descriptions below include assessments on whether each sea level index point is considered early or late 5e, which are separately denoted in **Figure 2**.  All elevations are reported relative to present-day mean sea level.

### 3.1 Abaco Island

Hearty et al. (2007a) describe two distinct episodes of reef growth below + 3 m, separated by an erosional unconformity on that truncates the lower reef and the adjacent eolianite substrate.  The elevations of the surfaces of the upper and lower reef are estimated at 2.6 and 2.0 m ± 0.2 m (see Chutcharavan and Dutton, 2021).  Hearty et al. (2007a) also mention sea cliffs, notches, sea caves, and planation surfaces at higher elevations (+6 to 10 m) on Abaco Island that may also be sea level indicators from MIS 5e.  Due to the lack of age data and other information associating specific features with measured elevations, we have not

included these additional geomorphic features from Abaco Island in the WALIS database.  One closed-system U-Th age passes geochemical screening from the upper of the two reef units, with an age of 120.6 ± 0.5 ka (Chutcharavan and Dutton, 2021). Given the truncation of the lower reef and the lack of detailed taxonomic or assemblage information on the corals, the position



of sea level relative to the observed surface remains uncertain but this surface can at least be considered as marine limiting. We have entered the lower reef as having a paleowater depth of a patch reef (indicative range of 0 to -6 m) without explicitly

adding an estimate of eroded material.

### 3.2 Andros Island

Two *Diploria strigosa* corals at Nicholls Town, Andros Island were dated to MIS 5e (120 and 128 ± 6 ka) using alpha-counting by Neumann and Moore (1975). These corals were reported at an elevation of +1.5 m relative to mean high tide, here corrected

to +1.9 m relative to mean sea level (adjusted by ½ of the tidal range at the Settlement Point tide gauge station: https://tidesandcurrents.noaa.gov/stationhome.html?id=9710441). It is not clear if the corals were collected from an *in situ* reef deposit; here we have applied a -3 ± 3 m paleowater depth, in keeping with a patch reef interpretation. A marine terrace at Nicholls Town was noted to terminate landward at the base of a cliff at +4.3 m relative to mean high tide by Neumann and Moore (1975), here adjusted to +4.7 m relative to mean sea level. These two estimates are consistent with each other, with the

coral yielding an estimated position of sea level at +4.9 ± 3.2 m compared to the estimate from the marine terrace at +4.7 ± 1.4 m. Although we have not included flank margin caves in the WALIS database entries, we note that several in this area occur at +5.0 m relative to mean sea level (Neumann and Moore, 1975).

### 3.3 Eleuthera

MIS 5e deposits are included from two locations on Eleuthera: Whale Point and Bioling Hole. At Whale Point, a section is described as containing poorly preserved corals that are presumed to date from MIS 5e (Hearty, 1998). Rovere et al. (2017) conducted differential global positioning system (DGPS) surveys in this area and report the highest *in situ* coral (*Diploria* sp.) at 4.42 ± 0.3 m and the surface of planar laminated beds at 6.71 ± 0.1 m. The laminated beds of which are interpreted to represent a paleo sea level position at 6.8 ± 0.6 m or 7.2 ± 1.2 m depending on whether this sedimentary feature represents

foreshore or beach deposits, respectively. In the WALIS database this is reported as 7.0 ± 1.0 m to capture these interpretations. Modern observations of *Diploria* spp. corals in the Bahamas range from -0.6 to -13.6 m relative to mean sea level (OBIS database, as reported in Hibbert et al., 2016). We have applied a typical patch reef paleowater depth estimate to the corals, which yields a sea level position of 7.4 ± 3.0 m.

Boiling Hole is the site of a stacked succession of two shallowing-upward sequences separated by an unconformity at ~ 4 m. This section has been correlated to the Grotto Beach Formation on the basis of whole-rock AAR ratios that are consistent with MIS 5e (Hearty, 1998; Hearty and Kaufman, 2000). Each sequence contains a transition from subtidal to beach facies, where the beach facies in the upper sequence is capped by an eolianite. The beach sediments contain fenestrae-rich, planar cross beds that are reported to occur between 2.5 and 3.3 m in the lower sequence and between 5 and 7 m in the upper sequence





(Kindler et al., 2010). Relying on the lowermost occurrence of the fenestrae (keystone vugs) as the most reliable position of past sea level then gives us two index points at 2.5 and 5 m. Since no datum for the elevation measurements is defined, we infer that the datum was the mean high tide mark based on the convention of many studies conducted in this region (e.g., Chen et al., 1991a; Hearty, 1998; Neumann and Moore, 1975). Adjusting these elevations relative to mean sea level yields the following sea-level positions: sea level at 2.9 m (lower MIS 5e unit), sea level at or below the erosional unconformity at 4.4

m, and sea level at 5.4 m (upper MIS 5e unit). Uncertainties of ± 1 m have been assigned to all these elevations in the absence of information about elevation measurement technique or uncertainty.

If we assume that the uppermost sea level indicators at Whale Point and Boiling Hole formed contemporaneously, the estimates of sea-level position at the top of each section do overlap if the uncertainties are considered (7.0 ± 1.0 m at Whale Point versus

5.4 ± 1.0 m at Boiling Hole). It could be argued that the 7.0 ± 1.0 m upper extent of the fenestrae-rich planar cross beds at Boiling Hole are a better match to the estimates from Whale Point than the lowermost occurrence that we have used here. We selected the lowermost occurrence of the fenestrae based on the logic that the higher elevations may represent the level of storm waves. Another possibility is that sea level was not stationary but rose over time to explain the 2-m vertical span of fenestrae-rich planar cross beds. If we were to use the uppermost occurrence of keystone vugs instead, then the three index

points from Boiling Hole would instead be sea level at 3.9 m, at or below 4.4, and then finally peaking at 7.0 m (with uncertainties of ± 1.0 m).

**3.4 Exuma Cays**

Jackson (2017) describes the MIS 5e deposits as common on the windward margin of the Exumas, displaying a subaerial exposure surface that is pink to brownish-red in color and featuring medium to large dissolution features (cm to meter scale).

The tops of MIS 5e eolian dunes are exposed, exhibiting orientations parallel (roughly N-S) and perpendicular (approximately E-W) to the windward margin. In several areas, multiple dune ridges are observed parallel to each other (and to the windward platform margin). Because carbonate dunes cement quite quickly after deposition, these multiple dune ridges were interpreted by Jackson (2017) as representing multiple positions of sea level during the MIS 5e highstand. Coral reef terraces were identified outcropping at about 1 to 1.5 m above sea level on the leeward side of some of the Cays, such as Rocky Dundas.

Several flank margin caves that occur in MIS 9/11 host rock occur in the Exumas that may date to MIS 5e, but lack age control and no elevations were reported in Jackson (2017). Here we have identified several sea-level index points, including a coral patch reef that yields an open-system model age of ~122 ka at an elevation of 1 ± 1 m at Norman's Pond, yielding a sea-level position of 4 ± 1 m. The age of these corals is uncertain owing to alteration (indicated by anomalously low initial $^{234}U/^{238}U$ activity ratios) but may correspond to the later portion of MIS 5e. The *Orbicella* sp. coral in a core from Little Darby Island

at -2.47 m suggests a sea level somewhere above this coral. In this case, we do not have the context of this corals to determine if it is from a patch reef or growing at greater depth. Again, the timing is uncertain owing to alteration of the coral (extremely high initial $^{234}U/^{238}U$ values) but based on open-system modeling may correspond to the later portion of MIS 5e also. The





beach ridges were interpreted as downstepping sea level at the close of MIS 5e. Cores that reveal the beach-to-dune transition record sea level down stepping from -8.5 to -9.2 to -10.2 m, although the timespan and absolute timing of this down stepping

is uncertain.

### 3.5 Great Inagua

The MIS 5e reef sequence at Devil's Point on Great Inagua has two well-defined phases of reef growth separated by a sharp, erosional unconformity that truncates the corals in the lower reef unit (Curran and White, 1995; Chen et al., 1991b; Hearty and Kindler, 1993; Hearty et al., 2007b; Skrivanek et al., 2018; Thompson et al., 2011; White et al., 1998). The unconformity is

flat on the promontories and slopes seaward in the embayments. It is also overlain in places by extensive coral rubble. The upper reef nucleates growth on the erosional unconformity and the coral rubble surface and the highest corals in primary growth position at Devil's Point display a lateral growth morphology indicative of growth in limited accommodation space. The corals in primary growth position that pass the flexible screening criteria in Chutcharavan and Dutton (2021) range in age from 128.3 ± 0.5 to 119.3 ka. The timing of the erosional unconformity estimated at between 124.5 ± 1.0 to 123 ± 0.6 ka

(Skrivanek et al., 2018). The truncated nature of the lower reef unit yields a higher uncertainty on the interpreted position of sea level, which was estimated at somewhere between 2.1 to 6.6 m for the lower reef unit (with the eroded reef surface at 1.1 m) and close to 2.1 m for the upper reef (which has the highest coral in primary growth position at 1.94 m). Hence the three sea level index points for this site are: Lower Reef: 2.1 + 4.5/-1 m at ~124.5 ka; unconformity: sea level at or below 0 ± 0.3 m at ~124 ka; and Upper Reef: 2.1 +3.8/-0.2 m at ~119 ka (see Skrivanek et al., 2018 for more details on uncertainties of coral

paleowater depths and ages).

### 3.6 Mayaguana

U-Th dates from a coral framestone unit on Mayaguana and amino acid racemization data from reef, eolian, and subtidal deposits yield ages corresponding to MIS 5e (Kindler et al. 2011; Godefroid et al., 2019). Three vertically stacked facies comprise the MIS 5e reef unit described near Misery Point by Godefroid et al. (2019) where the lowermost facies (0 to +1 m)

is predominantly massive *Pseudodiploria strigosa* framestone (122.5 ± 0.3 ka) encrusted by cm-thick red-algal crusts (123.2 ± 0.4 ka) (ages as recalculated by Chutcharavan and Dutton, 2021). This is superimposed by a ~1-m thick (+1 to +2 m) facies of encrusting corals and crustose coralline algae that is interpreted as a change in energy levels possibly due to shallowing (drop in sea level) (Godefroid et al., 2019). The uppermost facies extends to +2.5 m and marks a return to *P. clivosa* and *P. strigosa* (no age constraints, but interpreted as MIS 5e by stratigraphic correlation).

The paleowater depth interpretation of these reefs is not clear, although the vertical succession suggests a possible meter-scale sea level fall during MIS 5e based on the vertical transition in reef facies (Godefroid et al., 2019). The description of the lowermost coral assemblage is consistent with a 0.5 to 5 m paleowater depth interpretation, as described in Skrivanek et al. (2018). Godefroid et al. (2019) suggest a possible 5-10 m paleowater depth on the basis of Braithwaite (2016) but also point out that the assemblage of the terraces is typical of a moderate energy reef crest, which is more consistent with our



shallower paleowater depth assignment, and ultimately they assign a paleowater depth of 5-8 m (Table 5, Godefroid et al.,
2019). Here we consider the lower reef to have a likely paleowater depth of 0.5 to 5 m below mean sea level, and we estimate
the encrusting coral sequence to be shallower, perhaps within 1 m of sea level. Given the similar description of the uppermost
facies composed of monogeneric *Pseudoiploria* sp. to that observed on Great Inagua, assign a 0.2 to 3 m paleowater depth for
consistency to the paleowater depth interpretations for this assemblage (see Skrivanek et al., 2018).

These observations yield three sea level index points for the upper surface of each facies: +1.5 to +6 m for the lowermost
facies, +2 to +3 m for the middle encrusting coral facies, and +2.7 to +5.5 m for the uppermost monogeneric *Pseudoiploria*
sp. facies. If we accept the interpretation that sea level fell slightly in the middle facies, then this would further constrain the
lowermost unit to correspond to a sea level between +3 and +6 m. The timing of reef growth is constrained by three U-series
ages that show evidence of some coral alteration based on somewhat elevated initial $^{234}U/^{238}U$ ratios, but yield closed-system
ages in the range of ~125-122 ka, in broad agreement with other coral reefs across the Bahamas.

### 3.7 New Providence Island

MIS 5e deposits on New Providence Island have been the subject of numerous studies (e.g., Garrett and Gould, 1984; Aurell
et al., 1995; Hearty and Kindler, 1997; Reid, 2010; Jackson, 2017). The oolitic eolianites have been dated by U-Th (Muhs et
al., 2020) and AAR (Hearty and Kindler, 1997), confirming a MIS 5e age but lacking precision in terms of the exact timing
within the Last Interglacial owing to the challenge of recovering closed-system U-Th ages from ooids of this age. There is
also evidence of a subaerial exposure in the form of observed protosol and/or calcretes separating two submarine depositional
phases which have a small age separation in terms of the AAR data (Hearty and Kindler, 1997).

Jackson (2017) describes beach ridges with keystone vug horizons marking the position of sea level initially at +5.6 m to +5.0
(early MIS 5e), then falling below +1.7 m as evidenced by calcretes exposed at elevations ranging from +1.7 to +5.3 m that
separate the early (lower) MIS 5e from the later (upper) MIS 5e deposits. This brief regression is followed by a sea-level rise
to +6.0 m that then down steps to +4.5 and +3.5 m prior to glacial inception. Additional constraints come from a patch reef at
Northwest Point that has an elevation of +2.4 ± 0.1 m and returned U-Th ages between ~128 and 125 ka, corresponding to the
lower MIS 5e depositional record (Jackson, 2017; Muhs et al., 2020). Patch reefs in this region are often assumed to have a
paleowater depth of ~ 3 m (e.g., Chen et al., 1991), yielding an interpreted sea-level position of +5.4 ± 3 m, which is in
agreement with the inferred sea-level position from the keystone vugs (+5.0 to 5.6 m). Another patch reef is found eastward
at Gambier at an elevation of +3.8 ± 0.5 m represented by a coral assemblage of *A. cervicornis*, *Orbicella* sp., and *Diploria* sp.
that return ages of ~118-125 ka (Muhs et al., 2020). A single coral at Winton, another patch reef site that reaches +4.1 ± 0.5
m, was dated to ~118 ka (Muhs et al., 2020). Hence these two patch reefs seem to correspond to a later period of growth than
the patch reef at Northwest Point. Applying a 3-m water depth as before translates to a sea level position of up to +6.8 to 7.1
(± 3) m during this latter phase of reef growth, which may correspond temporally to the +6.0 ± 0.4 m sea-level position inferred
from the keystone vugs within the beach ridge at Clifton Pier.

While uncertainty in the paleowater depth of corals can introduce significant uncertainty into coral-based sea-level
reconstructions, in this case the agreement between the inferred position of sea level from the beach ridges and the patch reefs
lends confidence to the sea-level index points from New Providence Island. Ultimately, the MIS 5e deposits yield three phases
of sea level: the early or lower MIS 5e transgression that peaks at +5.6 ± 0.4 m (WALIS ID 3555), followed by an ephemeral
fall in sea level below +1.7 ± 0.1 m (WALIS ID 4128), followed by a rise in sea level to +6.0 ± 0.4 m (WALIS ID 4096) that
then down steps by a few meters before dropping more rapidly with the onset of glacial inception. The first sea-level highstand
presumably occurs between ~128-125 ka by correlation to the patch reef ages at Northwest Point, whereas the second sea-
level highstand represented by the corals at Gambier and Winton occurred between ~118-125 ka. The uncertainty in each of
these age windows is estimated at ~1 ka or possibly more, depending on the extent that open-system behavior has modified
coral geochemistry.

### 3.8 San Salvador

The reef exposures at Cockburn Town reveal a complex sea level history similar to that observed at Great Inagua with at least
two well-defined reef units separated by an undulating unconformity. Additional reef deposits are exposed at Sue Point and
Grotto Beach, and extensive MIS 5e deposits are observed across the island, including subtidal to subaerial features. The
analysis provided in Skrivanek et al. (2018) based on the Cockburn Town Reef provides us with three sea level index points:
the early/lower MIS 5e transgression is represented by the lower reef unit, with an estimated age of 128 – 125 ka based on U-
Th ages from the combined dataset from San Salvador and Great Inagua (Chen et al., 1991, Thompson et al., 2011). The lower
reef unit has an erosional upper surface with a maximum elevation of +1.5 ± 0.2 m and an inferred paleowater depth range of
0.2 to 3 m. Accounting for some erosion this corresponds to a sea level position of +2.2 to 5.0 ± 0.2 m. The +1.5 m erosional
surface provides a terrestrial-limiting data point representing an ephemeral sea level fall (magnitude uncertain) at ~ 124 ka.
This is followed by a rise in sea level represented by the upper reef unit to a position of at least +2.8 ± 0.2 m (reef surface) but
up to +4.1 ± 0.2 m according to the paleowater depth interpretation (see analysis in Skrivanek et al., 2018).

At Sue Point, just north of the Cockburn Town site, maximum elevation of reef surface is +1.4 ± 0.5 m relative to mean sea
level (based on reported elevation measurements from Chen et al., 1991, corrected from a datum of mean high tide to mean
sea level). Three ages that pass the strict screening criteria in Chutcharavan and Dutton (2021) indicate an age of ~ 122-123
ka that would correspond to the higher reef unit at Cockburn Town (later stage of MIS 5e). Applying a 3-m paleowater depth
correction (with a possible range of 0 to -6 m) for this patch reef facies gives us +4.4 ± 3.2 m for the interpreted sea-level
position, which overlaps the estimate from either the upper or lower reef at Cockburn Town.

The type section for the reef bearing MIS 5e unit in the Bahamas is at Grotto Beach on San Salvador Island. This section is
described in detail by Hattin and Warren (1989) who describe a reef unit containing *Porites asteroides*, *Psuedodiploria* sp.



and *Montastrea* sp. reaching up to +2.5 m, capped by a 10-20-cm thick layer of coralline red algae, *Neogoniolithon strictum*. The coralline red algae live in the uppermost subtidal to the lowermost intertidal zone and extend the coralalgal reef framework into the intertidal zone because of their ability to withstand subaerial exposure at low tide. Hence, this indicates that this patch reef built up to the uppermost subtidal. This agrees well with the shallow paleowater depth interpretation of the Cockburn
Town reefs (see above). At the Grotto Beach section, this coralgal unit is overlain by subtidal facies including some additional coral growth, although the elevation and description of this reef unit is vague. Upsection, this grades into a 'deeper' subtidal facies of 3-4 m paleowater depth, which, in turn, shallows upwards into 'shallow' subtidal facies (< 1 m paleowater depth) and then finally into parallel, seaward dipping beach facies that reach up to an elevation of +5.8 m. Hattin and Warren (1989) interpret the final position of sea level to be slightly lower, ~5 m, which agrees well with our estimation of the elevation beach-
to-dune transition relative to mean sea level (see Table 3). Hence, the Grotto Beach section demonstrates a paleo sea level position of +3.1 ± 0.4 m based on the algal cap in the lower (older) part of the outcrop, followed by a sea level rise that peaks at +4.8 ± 1.2 m. A pebble and cobble lag that separates the algal reef cap and the overlying deeper subtidal facies noted by Hattin & Warren (1989) may correlate to the erosional unconformity observed farther along the coastline at the Cockburn Town reef.


## 4 Turks and Caicos sea level indicators

The Caicos Platform is located on the southern end of the Bahamian archipelago. The highest quality age and elevation data that constrain Last Interglacial sea level position from this region are from West Caicos, a ~23 km$^2$ island on the southwest corner of the Caicos Platform. MIS 5e shallow marine deposits were identified through a combination of mapping and
radiometric dating (Simo et al., 2008; Wanless and Dravis, 1989) that were later refined by Kerans et al. (2019) who used a combination of AAR, U-Th dating, and elevation mapping using lidar with drone-acquired, georeferenced imaging.

### 4.1 West Caicos

As with most of the other sites within the Bahamas, a multi-phase depositional model was proposed for the West Caicos MIS 5e deposits where the South Reef (lower reef) unit was terminated by an erosional surface, capped by a laterally extensive
coral rubble layer, which in turn is overlain by a second (upper reef) phase of reef growth known as the Boat Cove unit. The stratigraphy of these MIS 5e units is well-exposed on the west coast of the island. We report three well-constrained sea-level index points from the Kerans et al. (2019) study as follows:

(1)      Towards the south end of the exposure of the lower South Reef unit, a coralline algal-reef crest facies is preserved with no evidence of erosional truncation at an elevation of +3.75 ± 0.2 m and an interpreted paleo sea level position of +4.15
± 0.4 m. The age of this unit is estimated at 126.5 ka by Kerans et al. (2019) based on U-Th ages from partially calcitized corals. Due to the mineralogic alteration, these dates were rejected by Chutcharavan and Dutton (2021) according to their

screening criteria. Nonetheless, these existing age data point to a MIS 5e origin, likely within the early to middle part of the interglacial.

(2)     The next stage includes significant erosion that removes as much as 2.5-3 m of reef, with the amount of erosion

increasing northward along the outcrop exposure down to a level of $0.5 \pm 0.2$ m above present-day sea level. The erosive surface is overlain by an extensive coral rubble unit that was interpreted to form as a result of lowering of sea level.

(3)     The third sea-level index point is captured by the Boat Cove unit (upper reef) that overlies the South Reef unit and coral rubble layer and has a distinct coral assemblage (patch reefs of Montastrea sp. (dominant) and Diploria sp. (subordinate)) relative to the reef crest facies observed in the lower reef unit. This unit has an algal cap of Goniolithon sp., which helps to

define the sea level position at $+4.15 \pm 0.3$ m based on maximum elevation of the coralgal cap at $+3.75 \pm 0.2$ m. The age of this unit is estimated to be ~120.5 ka by Kerans et al. (2019) based on U-Th dating of corals that are also somewhat calcitized. Like the corals in the lower unit, these ages were rejected in Chutcharavan and Dutton (2021). Hence, we consider this age estimate uncertain, but likely corresponds to the later portion (after ~125 ka) of MIS 5e.

The final phase of sea-level change during MIS 5e at West Caicos is represented by the Northeast Ridges, a set of ooid dune

ridges that prograded north and east as they stepped down below present-day sea level. These features are confirmed to correlate to MIS 5e according to the AAR data of Kerans et al. (2019). This progression of downward stepping ooid ridges at the end of MIS 5e is reminiscent of a similar observation of down-stepping ooid dune ridges farther north in the Bahamas in both New Providence Island and also in the Exumas where this is observed to continue down within -5 to -10 m as captured in cores (Jackson, 2017). At West Caicos, these dune ridges occur at elevations from +4-4.5 m toward the westerly initiation

point, stepping down to +0.7 m and continuing to the east of the present-day shoreline below present-day sea level. While the stratigraphy of these dune ridges indicates a falling sea level, there are no available RSL indicators that can be used to infer the precise position of sea level during this transition, nor is there a precise indicator of the timing, so we have not included it as a separate sea-level indicator in the WALIS database. Existing data suggest that this down stepping event began after deposition of the upper reef, reflecting the final regressive phase of sea level during MIS 5e.

**5 Florida (Atlantic Coast) sea level indicators**

As in the Bahamas, there is an extensive area in Florida thought to be submerged during MIS 5e based on the occurrence of shallow marine sediments that have been dated or correlated to this interglacial sea level highstand (**Fig. 3**). Here we have compiled data along the east coast of the state, including the Florida Keys; MIS 5e sea level indicators from the west side of the state are included in the compilation of Gulf Coast data (Simms, 2021). In general, the uncertainty is greater in age and

elevation of past sea level in the north and decreases in the southernmost region where U-Th dating of shallow marine carbonates (corals and oolites) affords increased age accuracy and precision, and the sedimentary archives also provide tighter constraints on past sea level position.

Along the east coast of Florida, the Atlantic Coastal Ridge is a notable topographic feature associated with MIS 5. The
Anastasia Formation (FM) consists of interbedded quartz sands and coquina (limestone) and outcrops near the surface of the
Atlantic Coastal Ridge along most of its length – from St. Augustine (just south of Jacksonville on the coast) in the north to
southern Palm Beach County in the south. This formation is named for the type section on Anastasia Island, where the coquina
was quarried to build the Spanish fort, Castillo San Marcos, and other structures. The coquina itself is primarily composed of
primarily fragmented molluscan shells that have been abraded, with scattered whole shells and quartz sand cemented in a
calcareous matrix.

Near the southern extent of Palm Beach County, the Anastasia Formation grades into the Miami Limestone, which consists of
an oolitic facies to the east and a bryozoan facies that underlies this and extends farther to the west. Farther south, the Miami
Limestone grades into the Key Largo Limestone, which is exposed in the northern Florida Keys and contains *in situ* coral reef
framework. Oolites from the Miami Limestone and corals from the Key Largo FM were initially dated by alpha counting
(e.g., Osmond et al., 1965) and have subsequently been dated using mass spectrometry in several studies that all confirm a
MIS 5e age for these units, although they demonstrate open-system behavior of the U and Th isotopes within these samples
often makes it difficult to determine the precise timing of deposition within the MIS 5e highstand (e.g., Fruijtier et al., 2000;
Muhs et al., 2011; Multer et al., 2002)


Though there are some specific elevation measurements for the occurrence of the Anastasia FM along the east coast of Florida,
the exact elevation of these deposits relative to contemporaneous sea-level position is debated. Clearly, the shells have
undergone some transport, and while the accumulations of fragmented shells in coquina deposits may represent shallow
offshore bars, shell accumulations can also occur in the supratidal zone. In general, it is also not clear what process led to the
concentration of fragmented shell material in this deposit (and the relative dearth of sand that is much more abundant in the
modern system), making the interpretations of sea-level position from this deposit uncertain. The age of the Anastasia FM
mostly relies on correlation to the laterally equivalent facies to the south that are better dated (see below). The Anastasia FM
has been dated using optically stimulated luminescence (OSL) (Burdette et al., 2009). Based on the elevation of the Anastasia
FM – where the coquina has been reported up to elevations of +6-8 m in places, it has typically been assumed to represent the
MIS 5e sea level highstand, although the bulk of OSL samples returned MIS 5c ages (Burdette et al., 2009). This agrees with
the OSL data from sand ridges at Merritt Island, Florida that also returned dominantly MIS 5c ages. Hence, there remain
questions about the timing of deposition of the Anastasia, which has been noted to include multiple phases of deposition (e.g.,
Randazzo and Jones, 1997). Given the considerable uncertainties surrounding the timing and indicative meaning of the
Anastasia FM, we have not included these outcrops in the WALIS database at this stage, though we note the widespread
attribution of this formation to MIS 5e along the east coast of Florida.

Muhs et al. (2011) provide an analysis of sea-level indicators in this southernmost region of the Florida Atlantic coast that provide the bulk of the sea level indicators for this region that we compiled for the WALIS database, and include both fossil corals and oolite deposits. U-Th dated corals from patch reefs containing abundant *Montastrea annularis* and *Diploria* sp. are

estimated to live in a paleowater depth of 3 m, though these corals can be found at both shallower and deeper elevations also. We note that the 3-m paleowater depth interpretation compares well with the interpreted position of sea level based on the elevation of contemporaneous oolitic shoals in the region. Florida oolites appear to have been deposited as shoals rather than the beach ridges observed in the Bahamas, hence their elevation relative to contemporaneous sea level may have been near sea level or as deep as ~3.3 m (Newell et al., 1960). We have included an estimate paleowater depth range of up to 3 m relative

to these ooid deposits to reflect this depositional environment.

We selected the highest sea level marker for each island or site to include in the WALIS database, as shown in **Figure 4**. In the northern keys and Miami region MIS 5e sea level is estimated at a peak level of +8.3 to 8.6 m, with somewhat lower elevations inferred from sea level indicators closer to ~ +5-6 m in the middle to lower keys. In terms of timing, many of the

coral U-Th measurements indicate open-system behavior but based on the array of data appear to record ages consistent with growth during the latter portion of MIS 5e (e.g., after ~123 ka). Despite open-system alteration in many measured samples, there are a handful of U-Th ages that pass a set of 'strict' screening criteria and 10 that pass a slightly more flexible set of diagenetic screening criteria (Chutcharavan and Dutton, 2021), which lends confidence to the interpretation that the dated deposits correspond to growth in the latter portion of the interglacial period. The sites included in the WALIS database are

described in more detail in Muhs et al. (2011).

Although there is some evidence for multiple peaks in sea level in the Florida MIS 5e depositional record (e.g., Usdun, 2014), other studies find no convincing evidence in terms of geomorphic, stratigraphic or geochronologic observations (e.g., Muhs et al., 2011).


## 6 Additional uncertainties

As in the Bahamas, there is an extensive area in Florida thought to be submerged during MIS 5e based on the occurrence of shallow

We have compiled a suite of sea level indicators across the Bahamas-Turks and Caicos-Florida region that provide useful

information about the past position of sea level during MIS 5e. The descriptions of the sea level indicators and data from the individual sites provided here also include some discussion of uncertainties that are specific to certain outcrops or types of indicators.

There are several additional sources of uncertainty that deserve a brief discussion. First, with respect to the reported elevations,
many studies do not even specify the method used to measure or estimate elevation, let alone the uncertainty associated with this measurement. We have used our knowledge of techniques often used in combination with the relatively small tidal range to assign some uncertainty to elevations where such information is lacking. Many authors in the Bahamas, for example, use the color changes in the algae growing on the rocky coastline to estimate the position of mean high tide and then estimate or measure elevations relative to that position. Many of these sites would benefit from modern surveying techniques to place all
the elevation measurements within a common reference frame.

Second, although WALIS quantifies the total uncertainty by adding the contributing errors in quadrature (e.g., uncertainty associated with elevation measurement combined with the uncertainty associated with the total indicative range of a sea level indicator), this assumes that the assignment of both the measurement uncertainty and the indicative range are known quantities.
In many cases, there exist differing interpretations of the relative water position of some of these sea level indicators. In general, we have included the most likely indicative range that encompasses any variability in interpretation that exists in the literature, so the reported uncertainty should be fairly conservative.

Third, we have provided the position of RSL as inferred from the elevation of sea level indicators across the region, but these
elevations also reflect the influence of glacial isostatic adjustment (GIA) across the region. These sites are positioned on the far flank of the peripheral bulge of the MIS 6 and MIS 2 Laurentide ice sheet, both of which significantly influence the modern observed position of these sea level indicators (Lambeck et al., 2012). As expected, there is a slight gradient in the elevation of the sea level indicators (higher closer to the former margins of the ice sheet and lower elevations farther from the ice sheet) due to GIA (e.g., Dutton and Lambeck, 2012).

Fourth, long-term subsidence has long been recognized in the Bahamas, and while some authors choose to apply subsidence rates on the order of ~2 meters/100,000 years to MIS 5e sea level indicators across the entire Bahamas platform, there is evidence that there may be significant differences in subsidence rates across the platform (Mullins and Lynts, 1977).

Finally, and related to the last point, there is ongoing uncertainty about the impact of dynamic topography on long-term rates of vertical motion across Florida and the Bahamian platform. It is important to note that we have not made any adjustments to the elevations of the indicators to account for any of these physical processes. However, they must be considered in the context of any future interpretations of sea level position or rates of change based on these data.

**7 Data availability**

The WALIS database is open access, and this data compilation can be updated and amended as necessary as it is not static. The data we have compiled and discussed here are available at (http://doi.org/10.5281/zenodo.5596898) (Dutton et al., 2021).

A description of each field in the database is contained at this link: https://doi.org/10.5281/zenodo.3961543 (Rovere et al.,
2020). Additional information on the World Atlas of Last Interglacial Shorelines can be found here:
https://warmcoasts.eu/world-atlas.html . We encourage users of our compiled data to cite the original sources alongside the
database and this article.

**8 Summary**

One important takeaway from this compilation is that we observe a widespread regional pattern of multiple peaks in sea level
that would complicate a straightforward interpretation of these sea-level data. Therefore, understanding the relative timing of
the indicator (early versus late MIS 5e) is critical for compiling data and comparing elevations across different sites. Our
compilation of data for the WALIS database demonstrates that reconstructions of sea-level change for this region should also
account for the variability of sea-level position across the duration of MIS 5e, including a brief sea-level regression that is
clearly imprinted in the sedimentary record.

The standardized MIS 5e database within this and other WALIS regional data compilations will provide a valuable tool for
future research efforts to constrain the behavior of sea level on glacial-interglacial timescales and to better characterize the
volume and extent of former land-based ice. While the metadata included in the database does an excellent job of quantifying
the uncertainties and quality of data, there will always exist nuances in the data that are hard to capture in the database format.
This is where the companion papers describing the regional WALIS compilations, such as this one, in addition to the original
data source publications cited herein can add value for future researchers who use these data in future studies.

**Author contributions**

AD identified data to be compiled; all authors contributed to data curation and interpretation; AV and PC entered the data into
the WALIS database; AD wrote the manuscript with contributions from all co-authors. Funding acquisition and project
supervision was provided by AD.

**Competing Interests**

The authors declare that they have no conflict of interest.

**Acknowledgements**
We acknowledge support provided by NSF awards #1702740 and #1443037 as well as graduate student support from the
University of Wisconsin-Madison gift funds. The authors acknowledge PALSEA, a working group of the International Union
for Quaternary Sciences (INQUA) and Past Global Changes (PAGES), which in turn received support from the Swiss Academy
of Sciences and the Chinese Academy of Sciences.



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






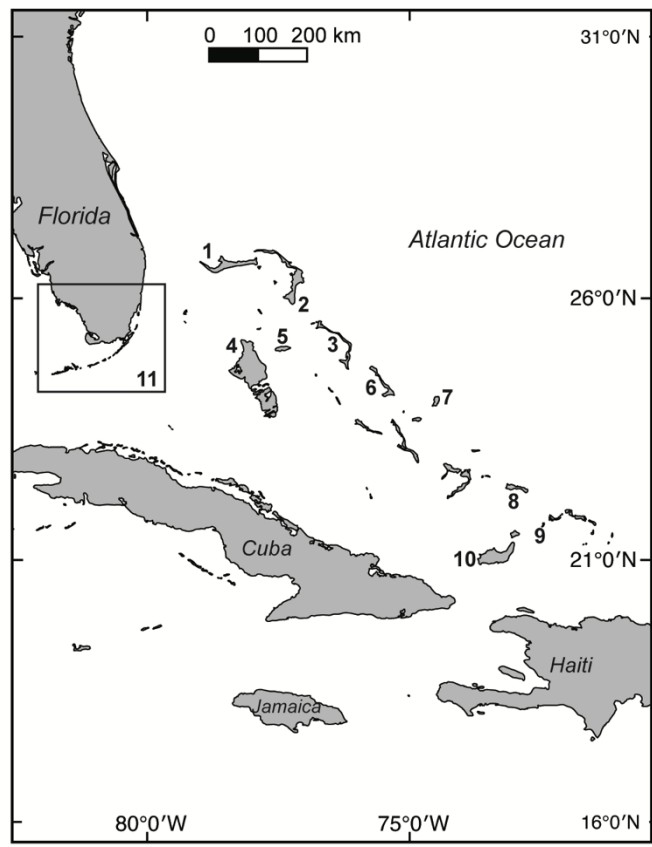

**Figure 1: Map of region considered in this data compilation, including the Bahamas, Turks and Caicos, and the east (Atlantic) coast of Florida.** Site numbers are as follows: (1) Grand Bahama; (2) Abaco; (3) Eleuthera; (4) Andros; (5) New Providence; (6) Exumas; (7) San Salvador; (8) Mayaguana; (9) West Caicos; (10) Great Inagua; (11) Southeast Florida region.


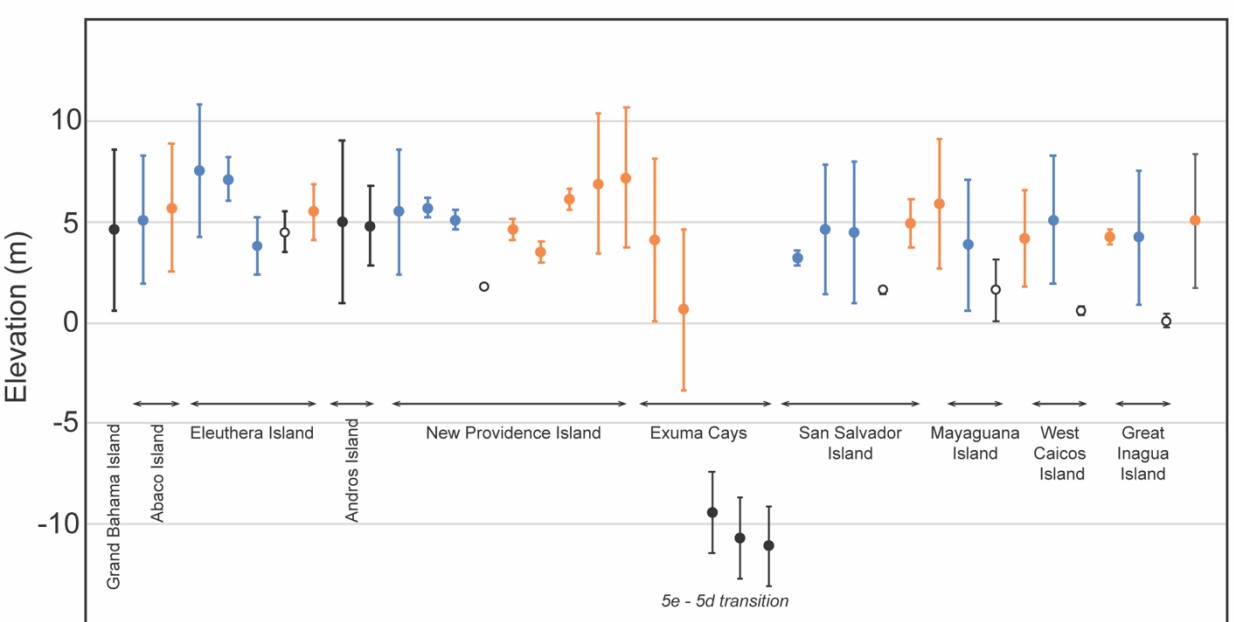

**Figure 2.** Relative sea level position during MIS 5e across the Bahamas and Turks and Caicos. Data are grouped by island and ordered roughly north to south from left to right across the diagram. Symbols are as follows (black circles = MIS 5e; blue circles = early MIS 5e; orange circles = late MIS 5e; open circles = erosional or subaerial exposure surface. Uncertainties represented by error bars include the combined uncertainty of the elevation measurement and indicative range of the sea level indicator by adding the uncertainties in quadrature.

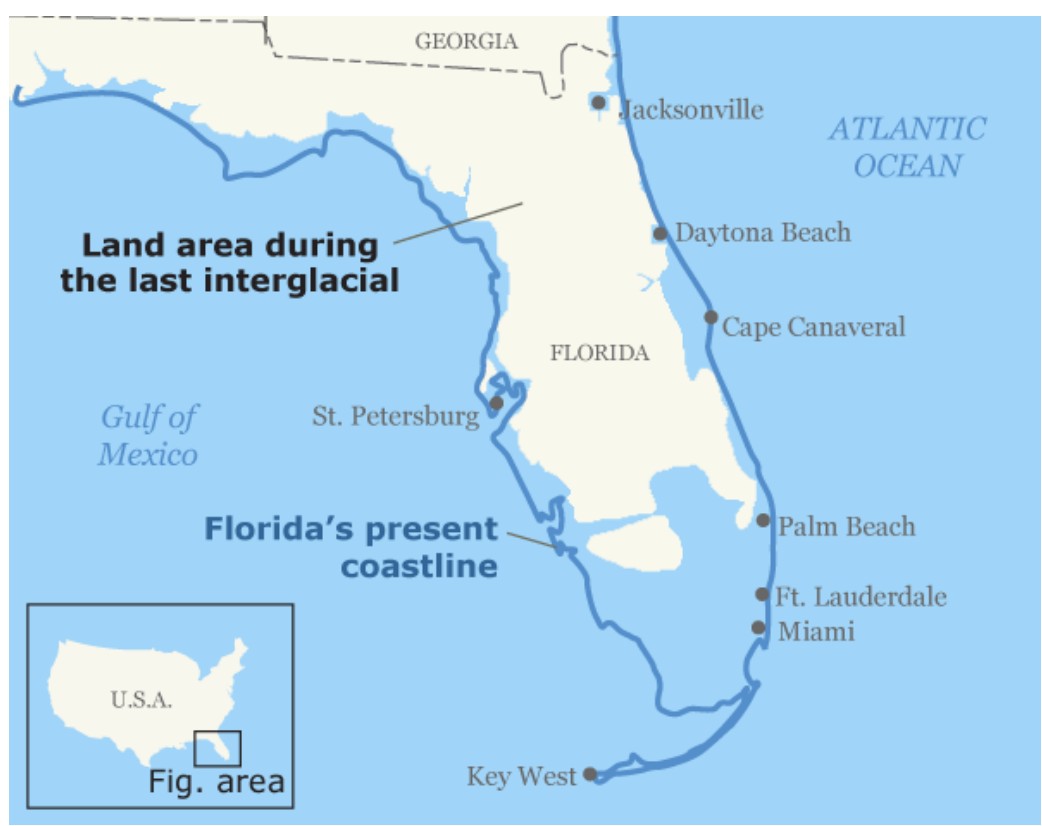

**Figure 3.** Florida's Last Interglacial shoreline versus that of today. Solid blue line denotes present-day shoreline relative to the position of the interpreted position of the Last Interglacial (MIS 5e) shoreline that is shown by the boundary between the white (land) and light blue (ocean) shaded areas. (USGS/public domain).





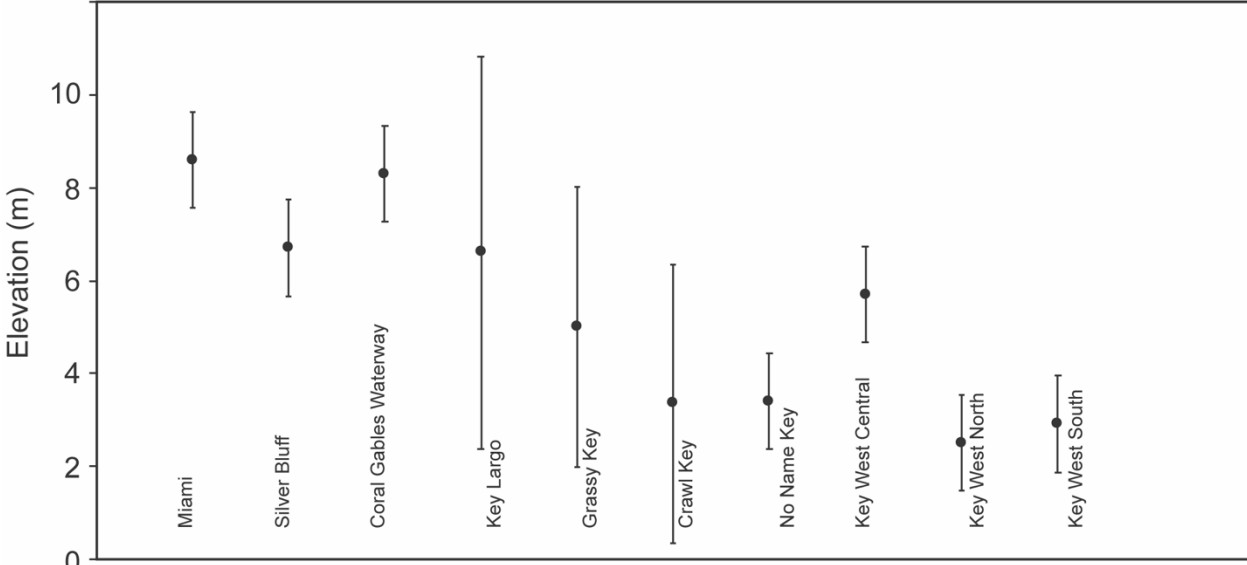

**Figure 4.** Relative sea level position during MIS 5e along the east coast of Florida, with sites plotted by decreasing latitude
(north to south) from left to right across the diagram. Uncertainties represented by error bars include the combined uncertainty
of the elevation measurement and indicative range of the sea level indicator by adding the uncertainties in quadrature.