# Peer review of "Compilation of Last Interglacial (Marine Isotope Stage 5e) sea level indicators in the Bahamas, Turks and Caicos, and the east coast of Florida, USA"

_Earth System Science Data, 2021_

## Referee Comment (RC1)

Review of Data Synthesis paper – Compilation of Last Interglacial (Marine Isotope Stage 5e) sea level indictators in the Bahamas, Turks and Caicos, and the east coast of Florida, USA - Andrea Dutton, Alexandra Villa, and Peter M. Chutcharavan

Following the 4 questions put forth by Earth Systems Science Data

1. *Is the article itself appropriate to support the publication of a data set?* Yes, this paper is clearly written, provides a good explanation of the parameters by which SL indicators are assessed and errors/uncertainties documented, and provides a useful compilation of cross Bahamas/Caicos/Florida RSL data in a standardized fashion along with many of the key references.
2. *Is the data set significant – unique, useful, and complete*? – The data set is significant, and unique in the sense that normalized data across this key region are provided in one location. Is it complete? Is any data set complete? To the extent possible it is, yes. Some references could be updated.
3. *Is the data set itself of high quality?* – Yes, significant effort was made to ensure that data were considered in the context of a guiding set of norms, which ensures data quality.
4. *Is the data set publication, as submitted, of high quality?* – The data set pub is of high quality for what it is. The writing is clear and concise, and provides very useful summaries of multiple localities and recent studies.

As a final measure, we are asked, would you be able to understand and (re-)use the data set in the future. Absolutely, if fact I plan to do so in the near term for my own research.

**General Comments -**

The paper does a good job of highlighting the multiple examples across the study region where multiple peaks of RSL are recognized from MIS 5e.

I find the tables discussing quality of SL elevation and age constraints clear and useful. A few comments on these below for consideration

Flank margin caves – I agree totally that it is a good idea to not include flank margin caves in the database. You might add the additional new ref that discusses and challenges the Mylroie model for process of formation of FMCs

Breithaupt, C.I., Gulley, J.D., Moore, P.J., Fullmer, S.M., Kerans, C. &
Mejia, J.Z. (2021) Flank margin caves can connect to
regionally extensive touching vug networks before burial: Implications
for cave formation and fluid flow. Earth Surface Processes
and Landforms, 46(8), 1458–1481. https://doi.org/10.1002/esp.
5114

Keystone vugs –some cautionary note to say that some occurrences of keystone vugs can be substantially higher than the -0.4 to +0.4 m mentioned here, likely associated with storm-surgegenerated swash. An example of anomalously elevated keystone textures could come from Wanless and Dravis (1989) p. __ where keystones occur at +__.

Patch Reef vs Fringing Reef distinction - I would suggest that several examples of reef systems observed (at least West Caicos and I suspect Cockburn Town as well) are more like a fringing reef, growing right onto/attached to a 9/11 substrate that is seaward-dipping. Not sure it matters, but both reefs have well-developed coralline-algal caps (as does the little reef occurrence at Grotto Beach). Most patch reefs I have investigated do not have coralline-algal caps, but are as you suggest, -3 m water depth range.

Devil's Pt, Great Inagua - Without seeing the outcrops, I would speculate that the lower 5e Devils Pt reef which is a mix of A. palmata, Orbicella (M. annularis) and Diploria is a truncated erosional remnant of a land-attached fringing reef (presumably on MIS9/11 strata) that would likely have had a coralline algal cap. Just speculation at this point though. Of course the significance here is that the -3m estimate provided.

Wave-cut notches – these have been used by previous authors, and though clearly less diagnostic, and not possible to date, should at least be mentioned for completeness.

Oolite shoals – best reference for average depth of formation of different styles of ooid grainstone might be from Purkis and Harris (2017) stating that "the relief of the contemporary Great Bahama Bank ooid sand bodies spans -2 - -10 m beneath present day sea level". However, the detailed analysis of Purkis and Harris 2017 on the Miami oolite suggests that these sand bodies extended from SL to -2 or -4. So a range might be good. But at any rate these refs would be more up to date and quantitative for SL analysis. Got a confirmation from Mitch Harris that the -2 to -4 m range for ooid formation was in agreement with most published data.

**Comments on information from specific regions -**

Bahamas Sea Level Indicators – 1$^{st}$ paragraph

I am not sure how to best handle this. This is a consistent rendition of the published literature on the San Salvador island 5e stratigraphy, but this stratigraphy is demonstrably incorrect. I would recommend keeping this brief so that it does not "date" or limit the work at hand. It would be accurate to say that Carew and Mylroie (1985) proposed the two-member division of the 5e with the French Bay and Cockburn Town members. It was not ever demonstrated that the Cockburn Town member is volumetrically more abundant, or that the French Bay stratigraphically underlies the Cockburn Town, so I think it would be better to leave this out.

Also, you mention Titus (1980) here, though all the cited refs for Titus seem to be 1981 or 1982 and are from the grey literature of the Gerace Research Lab proceedings which are not properly refereed or published, so I think you could just leave it to the C and M 1985 and 1995 refs.

I would also suggest that the Cockburn Town member is a challenging unit in that it cannot be demonstrated as to which of the 5e SL peaks it is associated as there are both lower and upper 5e Cockburn Town reefs. By far the most abundant facies in the 5e in San Salvador and on other Bahamian islands are associated with the upper MIS 5e and are predominantly oolitic beach-dune strata, as demonstrated for West Caicos.

Abaco – good summary of what is available. I would mention the Hole in the Wall locality as I believe that this is where all this Abaco data come from. Makes sense to me that the notches are mentioned, but not included in the database as their age/significance is really difficult to tie down.

Andros Island - Coral ages from Neumann and Moore 75 are essential. Newer work in the area adds a lot to the sedimentology, but not to understanding of SL position (eg. Hazard et al 2017 JSR)

Eleuthera – Good summary of data given from the more rigorous Whale Pt and more general Boiling Hole. However, importance of Boiling Hole is clearly the presence of double 5e record w unconformity. From material presented without petrography, I suspect the aeolian strata illustrated in Rovere et al (2017) are in fact MIS 5a strata, not MIS 5e, so the foreshore to dune transition value might not be valid. Also, the authors do not mention that rather than a "reef" here at Whale Pt, the coral heads are individuals resting unconformably on older mid-Pleistocene bedrock.

Exuma Cays – The description of multiple parallel dune ridges I believe could be left out here as the only true indicator of SL is the dune to FS transition. Different dune heights could tie to SL or could tie to changing sediment supply or climate/vegetation, so do not know if this info should be included. The distinctly negative SL position for the dune ridges is interesting, representing the lowest elevation dune to FS transitions I am aware of, but unless they fit into a broader framework, it is hard to place these.

Great Inagua – Reading this description it seems that GI is placed at equal confidence with other sites such as Exumas or Eleuthera, but I believe that with the extensive age control, the very clear unconformity, and the two superimposed reefs, that somehow this site needs to be noted as a key "must see" site for understanding Bahamas MIS 5e SL. The SL positions and how they are assessed here might be better addressed as it is somewhat unclear how the SL of the lower reef was made. The statement "*The truncated nature of the lower reef unit yields a higher uncertainty on the interpreted position of sea level, which was estimated at somewhere between 2.1 to 6.6 m for the lower reef unit (with the eroded reef surface at 1.1 m)*" I am gathering that the 4 m of uncertainty here stems from the range of coral growth depths and not from the uncertainty tied to the amount of erosion, am I getting that correct?

Mayaguana – Again useful broad constraints and ages, but not sufficient to add to the multi SL peak framework as discussed. An age range of 125-122 ka and the RSL positions of +1.5- +6 and +2.7 to +5.5 are in broad agreement with data across the region.

New Providence -  Good, captures all most recent info

San Salvador -  Cockburn Town summary is good and well documented by the Skrivanek et al paper plus all previous age work. When mentioning Sue Pt, There is also an upper 5e reef with near identical elevation and facies relationships at Hall's Landing, in addition to Grotto Beach and then on the south coast at the Bluff, and also a small patch reef just west and south of Crab Cay. Hall's Landing, Sue Pt, and Grotto all, on the basis of gradational upward contacts with upper MIS 5e grainstones, must fall in the upper MIS 5e grouping. The Hattin description is somewhat bogus. They forget to mention that the patch reef there is growing on Owls Hole erosional remnant, and then grades up through corals into dominantly coralline algal cap and then foreshore and ultimately dune. We have the dune fs transition at +5m, but at the same locality Dyer et al us a 5.8m number. That is published so perhaps best to go with that one.

West Caicos – In contrast to the statement on line 380, the patch reefs in the upper unit (Boat Cove unit) make up 2% of the unit, the rest is ooid gnstn, so better to suggest that basal transgressive deposits of the upper 5e Boat Cove unit are small local patch reefs, that are succeeded by voluminous ooid grainstone that makes up the bulk of the upper 5e.

WRT the lack of RSL is suggested for the NE ridges, in fact there are excellent foreshore to dune transitions in multiple localities. But not in the paper so fair enough.

Eastern margin of Florida – line 423-424  - use primarily twice

I am surprised by the high RSL values posted for northern Keys and Miami. In keeping with our examination of the lidar data as well as that published by Purkis and Harris (2018) a RSL value for the Miami oolite might be closer to +5-+6.

6 additional uncertainties – something is missing on line 478

In summary, this is a  high-quality publication that does exactly what it set out to do in documenting data on RSL indicators for the Bahamas-Caicos-Eastern Florida. It has done a good job of presenting data that is well documented and leaving out less certain information. Clearly a follow-up paper that provided a more in-depth look at the data sets and comparison across data sets with new illustrations and analyses would be highly desirable, but that is beyond the scope of this useful publication.

**Ratings** Uniqueness, 2; Usefulness, 1; Completeness, 1

Typos
Line 72 "elevation of a where a…"

---

## Author Response (AR1)

**Responses to Reviewer 1:**
We would like to thank Charlie Kerans for providing comments on this manuscript and for the constructive criticism.

Below we respond to the points raised:

(1) Note about reference to include on flank margin caves.
*This reference will be added.*
(2) Keystone vugs – cautionary note that some keystone vugs can be substantially higher than the range mentioned here (-0.4 to +0.4 m).
*Thank you for emphasizing this point. We agree, which is why we noted in section 2.5 that "the lowest occurrence of keystone vugs is typically taken as the best estimate for the position of sea level." In each of the sites, we followed this convention of using the lowest occurrence of keystone vugs to mark the position of sea level to avoid bias from keystone vugs formed during storm surges. We have clarified this by adding a sentence in section 2.5.*
(3) Patch reef versus fringing reef distinction is lacking
*Here the point is made that some of the reef outcrops may have different paleowater depths (not always -3 m) and may be fringing reefs with coralline-algal caps. We agree that -3 m is an estimate that we used where other information is lacking with respect to paleowater depth or extent and type of reef deposit based on existing reports. The sites mentioned in this comment do have more detailed analyses of paleowater depth (Grotto Beach, Devil's Point, Cockburn Town, and West Caicos). In all of those cases, we used the published information to further refine water depth at those sites. This approach was described in section 2.1 as follows: "Here we adopt an estimated paleowater depth (or RWL) of -3 m relative to mean sea level, with a possible depth range (IR) of 0 to -6 m for patch reefs in this region unless there is more specific information available that can be used to refine this range. "*
(4) Should mention wave-cut notches for completeness
*Good point. We have added section 2.7 on wave-cut notches.*
(5) Summary of Bahamian stratigraphy may not be entirely correct, despite the face that this is an accurate reflection of the published stratigraphy.
*We have modified this section to remove references to the relative position of the Cockburn Town and French Bay members, which is not clearly demonstrated in the literature.*
(6) Suggest to remove Titus, 1980 reference as it is from 'grey literature.'
*This has been done.*
(7) Suggest to include the name of 'Hole in the Wall' in Abaco island description.
*Good idea. This has been done.*
(8) An additional reference was noted that is relevant to Andros Island
*We have included a reference to the sedimentology in Hazard et al., 2017*
(9) Exuma Cays – comment that beach dune elevation does not represent sea level position
*We agree. Our wording was misleading here and we have modified the wording to clarify that the series of dune ridges was interpreted by Jackson (2017) to represent shoreline migration.*
(10)     Great Inagua – some confusion over why there is a larger uncertainty here
*We have added wording to clarify that the additional uncertainty for the lower reef unit here stems from having to estimate how much material was removed during erosion.*

(11)  West Caicos – comment to revise description of Boat Cove unit
   *Thank you for this clarification. This is done.*
(12)  Something missing on line 478
   *Thank you. Sentence fragment was removed.*
(13)  *Typo on line 72 corrected.*

**Responses to Reviewer 2:**
We would like to thank Blake Dyer for his constructive comments on this manuscript. Responses to specific comments are listed below.

[Comments in regular font, responses in italics]

The discussion of uncertainty in tidal datum comes late in the manuscript
*We have called out this section earlier in the manuscript now to alert the reader of these uncertainties before the individual sites are discussed.*

Suggest to include additional references for Cockburn Town U-series dating
*Good suggestion. Citations for Chen et al (1991) and Thompson et al (2011) have been added.*

Suggestion to clarify original data source
*We have added a citation to the original publication at the appropriate spot in this sentence to make this clarification*

Please make tidal adjustment more clear
*Done.*

Typo on Line 210.
*Corrected.*

Line 270: include Thompson et al., 2011 citation.
*This citation is included above. In this line we are describing an interpretation of the age data that was described and interpreted by Skrivanek et al., (2018) so this is the appropriate citation to explain where that interpretation comes from.*

Line 293 – Typo.
*Corrected, thank you.*

Section on New Providence Island is missing keystone vugs from Garrett and Gould that were at higher elevations.
*The elevations published by Garett and Gould (1984) were corrected for 3 m of subsidence and measured relative to a different datum. All elevations here are relative to mean sea level as noted at the beginning of Section 3. Once the G&G elevations are corrected, these match the elevations reported and re-surveyed by Jackson (2107). This example highlights the importance of compiling all of these sea level data within the same framework.*

Line 343- why is this erosional surface recorded as a terrestrial limiting point?
*In the following sentence, we referred the reader to the analysis in Skrivanek et al. (2018) which explains the sequence in more detail, including a justification for subaerial exposure. To make this connection more clear, we have combined these two sentences together so that the Skrivanek reference can provide the reader with more details on this as needed.*

Line 500 – Suggest to add a citation for Dyer et al. 2021
*This citation is now included.*

Line 508 – Not clear why this statement is attributed to Mullins and Lynts.
*The citation to Mullins and Lynts was removed.*